# Antipsychotic Use: Cross-Sectional Opinion Survey of Psychiatrists in India and United Kingdom

**DOI:** 10.3390/pharmacy11050162

**Published:** 2023-10-09

**Authors:** Zina Sherzad Qadir, Nilamadhab Kar, Patrick Anthony Ball, Hana Morrissey

**Affiliations:** 1Research Institute in Health Sciences, University of Wolverhampton, Wolverhampton WV1 1LY, UK; sherzad.zhina@gmail.com (Z.S.Q.); n.kar@nhs.net (N.K.); pball@csu.edu.au (P.A.B.); 2Black Country Healthcare NHS Foundation Trust, Wolverhampton B71 4NH, UK; 3School of Dentistry & Medical Sciences, Charles Sturt University, Wagga Wagga, NSW 2678, Australia

**Keywords:** antipsychotics, antipsychotics discontinuation rate, causes of relapse, optimisation of schizophrenia management, psychiatric disorders

## Abstract

The aim of this survey of psychiatrists from the UK and India was to compare their opinions on antipsychotic medication choice and their experiences of such medications’ effectiveness and tolerability in patients who were newly diagnosed with acute schizophrenia. Following ethical approval, a cross-sectional online survey of psychiatrists from the UK and India was conducted. Ninety-five responses were received from each country. The most selected first-line APDs in both countries were olanzapine (47.5%), risperidone (42.8%) and aripiprazole (25.3%). A total of 60% of psychiatrists from India (60%) and 48% from the UK (48%) selected ‘medication efficacy’ as the main factor in their choice. Reassessment and consideration to switch most often took place within 4–6 weeks (53.7%) and 3–6 months (11.6%). The major reasons for switching antipsychotic medications were poor clinical efficacy (69%) and lack of tolerability (45%). Nonadherence was the most common reason for relapse (90% of UK psychiatrists and 70% of Indian psychiatrists), followed by illicit drug use (27.6%). The most commonly reported side effects that led to nonadherence were weight gain (10.8%), drowsiness (10.4%), erectile dysfunction and movement disorders (equally 8.7%). It was concluded that olanzapine, risperidone and aripiprazole are the most commonly selected as the initial treatment choice by psychiatrists from India and the UK. They are perceived as widely effective and well tolerated.

## 1. Introduction

Approximately 220,000 people in the UK are currently receiving treatment for schizophrenia [1]. APDs are indicated as the first line of treatment for schizophrenia, coupled with psychological intervention for the individual and their families [2]. APDs are generally classified into typical APDs (first-generation, FGA) and atypical APDs (second-generation, SGA). The British National Formulary (2022) listed the APDs available to prescribe in the UK (Table 1) [3].

FGAs were developed in the 1950s and began with the discovery of chlorpromazine. FGAs are nonselective dopamine receptor blockers, limited by their high incidence of extrapyramidal side effects (EPS) [3]. Their lower cost compared to the SGAs has led to the continuation of their use, particularly in low-income countries [4]. They are ineffective in treating the negative symptoms of schizophrenia [3]. SGAs have more distinct side effect profiles, particularly with regard to EPS, which is derived from their moderate dopamine receptor blockade compared to the FGAs [3]. It has been hypothesised that serotonin (5-HT) neurotransmission is also disturbed in schizophrenia. SGAs also act on acetylcholine, histamine, norepinephrine and 5-HT receptors, which leads to a higher risk of weight gain and glucose intolerance [5]. Improvement of negative symptoms and cognition by SGAs is mainly due to their potent 5-HT receptor antagonism and relatively weaker dopamine antagonism [6]. The SGA clozapine improves positive and negative symptoms of schizophrenia and has the least EPS; however, due to the possible life-threatening agranulocytosis associated with clozapine, the use of this drug is only advised if frequent monitoring of white blood cells is possible [7].

There is evidence of broadly similar efficacy of all currently available APDs within their generation (FGA or SGA) [8]. However, there is a wide disparity in the individual level of response to treatment, symptom control and the experience or severity of side effects [2]. Clozapine is indicated in treatment-resistant schizophrenia (TRS) when a patient is deemed as unresponsive, or intolerant, to conventional APDs [2]. The UK Prescribing Observatory for Mental Health (POMH) reported a gradual change in the use of dual APDs and in the switching from one APD to another between 2006 and 2017 [9]. In 2006, 43% of patients in an acute adult department in the UK received more than one regular APD medication or switched APDs, which decreased to 32% in 2017. The main clinical reason for dual or alternative use of APDs is poor efficacy of monotherapy or as an overlap during APD switch [9]. Patients diagnosed with a schizophrenia-related first episode of psychosis (*n* = 50) completed a survey at South London and Maudsley NHS Foundation Trust to evaluate attitudes towards APD non-adherence. The authors found that most patients (66%) reported good adherence [10]. Non-adherence to APD regimens caused an increase in the annual cost of total care per patient of GBP 5000 due to relapse [11,12]. Additionally, the clinical records of 1515 patients with a schizophrenia spectrum disorder from four London boroughs indicated that 17% of all patients developed TRS [13].

Early identification is important in order to prevent longer hospitalisation and reduce the total burden on health services [14,15]. The relapse rate in patients with first-episode psychosis was reported by the Birmingham Early Intervention Service between 2012 and 2015. The Kaplan–Meier calculation for 12-month relapse rate was estimated at 67%. This demonstrates that relapse is common after APD discontinuation, following recovery from the first-episode psychosis. Moreover, the Service identified relapse predictors as: male gender, unemployment, those who were not in education or starting new training, those who were in new employment and those who have had previous hospital admissions [16].

There are important global similarities and differences in approaches to managing psychotic disorders. Two countries were included in this survey: the UK, as the researcher location, and India, as Indians represent the most populous Asian ethnic group residing in England and Wales (1,412,958 people from the Indian ethnic group, 2.5% of the total population as per 2011 census) [17,18]. The current model in both countries is based around acute management by specialists based in general hospitals, with specialist hospitals only for the most severe or treatment-resistant cases [19]. Traditional religious and cultural beliefs and stigma affect attitudes toward seeking and providing help, intentions to initiate treatment and the setting/period of treatment maintenance. This is seen to a greater extent within India and people of Indian descent living in the UK [19,20]. Mental health public services are under-resourced in both countries. The cost of treatment maintenance is largely absorbed by the National Health Services (NHS) compared to private care in the UK, whereas in India, the cost is shared between the government and the patient [20]. India is particularly short on consultant psychiatrists (0.3–1.2/100,000 people), with large variations between states [21,22]. The figure in the UK is 13/100,000, but also with significant variations by region [23].

## 2. Methods

The hypothesis developed for this survey was as follows: Though some APDs are more favourable than others, the optimal APD is one that suits the patient’s condition and which the patient can tolerate.

Interviews were deemed not suitable due to the difference in the time zone between the two countries and COVID-19 infection control restrictions in operation at the time. A web-based survey format was chosen [24] for its low cost and to accommodate psychiatrists’ busy practice schedules, allowing them to complete the survey at any convenient time. This survey was conducted to investigate real-world practice in the UK and India to gain a better understanding about practices related to the use of APDs in the treatment of psychosis.

In the UK, all psychiatrists were approached who had public social media accounts or a publicly published email address, and those with social media accounts only were approached in India. Additionally, Instagram^®^ and LinkedIn^®^ were used for recruitment from the UK and India, with a total of 750 psychiatrists invited. The survey was active for 3 months from 26 April 2022 to 31 July 2022. The expected total time taken by participants to complete the questionnaire was approximately 30 min. The survey invitation provided an access link to the participant information sheet, which made clear that the survey was completely anonymous; however, anonymity meant that once the ‘submit’ button was selected at the end of the survey, it was no longer possible to withdraw. A final consent ‘yes/no’ box then gave access to the survey questions. Submission of the completed survey was therefore accepted as implied consent. The survey was hosted on the Joint Information Systems Committee (JISC) Online© survey software application, which is approved for health research surveys. Only two reminders were sent, and the survey was left open for 10 days after the last response was received.

All responses were anonymous and consisted of 25 questions: 3 questions to collect demography data to facilitate the statistical analysis (gender, country of practice, practice setting) and 22 opinion questions (multiple options, closed and open-ended questions). Questions 5–10 concerned the prescribing of APDs, including the preferred agent for initial therapy and reasons for switching APDs. Questions 11–13 were about relapse and hospitalisation, and questions 14–16 focused on the psychiatrists’ preferred choice of formulation. Questions 17–22 explored perceptions around the most troublesome side effects of APDs leading to non-adherence or discontinuation. Question 23 looked at categorising APDs in order of their perceived efficacy, and question 24 was about the most common challenges faced when managing psychotic disorders. Finally, question 25 was an invitation for any further comments. The questionnaire was reviewed by a practicing psychiatrist. The study was reviewed and approved by the Life Sciences Ethics Committee (LSEC) of University of Wolverhampton (LSEC/202021/HM/8) on 13 April 22 and by two NHS Hospital Trusts for the distribution of the survey into their psychiatrists. It was then pre-tested with a small group of psychiatrists (two from each country) prior to dissemination.

The selection criteria included being a licenced psychiatrist, or resident in training, in the UK or India and having an active involvement in patient management and treatment decision making for patients diagnosed with schizophrenia. A total of 190 psychiatrists completed the survey questions, a response rate of 22%, which is at the lower end of response rates but not unusual [25,26,27]. Statistical analysis was carried out using IBM^®^ Statistical Package for Social Sciences (SPSS™) version 28.0.0.0. Survey responses were initially downloaded and saved into a secured Microsoft^®^ Excel™ spreadsheet. The UK spelling of the APDs’ names was used. Participants were allowed to select all options that applied for the multiple options questions (8, 10, 11, 12, 13, 14, 16, 17, 18, 19, 20 and 23). Statistical analysis or thematic analysis was conducted as applicable to the type of data. Thematic analysis was conducted in accordance with the Braun and Clarke [28] process of creating codes and themes. Categorical results were reported as the number of responses and the percentage of total responses to the individual question. Pearson Ꭓ^2^ was used to test for significance, with asymptotic *p* < 0.05 as significant.

## 3. Results

Frequencies and percentages were calculated for the relevant category, total responses, country, gender etc. The total sample (*n* = 190) was equally distributed between the two countries (*n* = 95, 50% from each). Of all responses, 72.6% (138/190) were males, and 25.8% (49/190) were females; three participants selected the ‘prefer not to disclose’ option (1.6%). Most participants (67/190, 35.3%) were employed in acute care psychiatric hospitals, followed by those employed in private practices (49/190, 25.8%), long-stay specialist hospitals (31/190, 16.3%) and community psychiatry practices (28/190, 14.7%). The remaining respondents (6/190, 3.2%) worked in general hospital psychiatric units, teaching hospital tertiary care centres (3/190, 2.1%), the Ministry of Defence (3/190, 1.6%) and one in the prison system (0.5%).

### 3.1. Prescribing Preferences

The most frequently selected APD as a first line for the treatment of acute psychosis was olanzapine (87/190, 47.5%), followed by risperidone (46/190, 25.1%), aripiprazole (41/190, 22.4%), quetiapine (4/190, 2.2%) and trifluoperazine (3/190, 1.6%), and the least selected was haloperidol (2/190, 1.1%). Olanzapine was the most selected by Indian practitioners (51/95, 54%), but in the UK, it was equally selected with aripiprazole (36/95, 38% each). Olanzapine was the highest selected in specialised and private hospitals (48/67, 48% and 25/49, 51%, respectively). The first choice of APD by country was significantly different (*p* < 0.001). The most reported second-choice agent was risperidone (68/190, 42.8%), followed by olanzapine (47/190, 29.6%), and the least reported were paliperidone, zuclopenthixol and amisulpride, which were each chosen by one only. Risperidone was the highest selected second option by female (15/49, 31%) and male (53/138, 38%) practitioners. It was also the most selected second-choice agent by Indian (42/95, 44%) and UK practitioners (26/95, 27%). Olanzapine was the highest selected fall-back in specialised and private hospitals (37/67, 37% and 21/49, 43%, respectively). The second choice of APDs between countries was also significantly different (*p* < 0.001).

The third most frequently prescribed APD was aripiprazole (39/190, 25.3%), followed by quetiapine (27/190, 17.5%), and the least selected was ziprasidone (1/190, 0.6%). Aripiprazole was the highest selected third option by males (31/138, 22%); however, it was equally selected with olanzapine among female (10/49, 20%) practitioners. Aripiprazole was the highest selected by Indian practitioners (24/95, 25%); however, olanzapine was the most selected by UK practitioners (21/95, 22%). Olanzapine was the highest selected in specialised and private hospitals (23/67, 23% and 11/49, 22%, respectively). The option of third-choice APDs by country was significantly different (*p* < 0.001).

Medication efficacy was selected as the first reason by Indian practitioners (90/95, 95%), where medication safety was the top choice for UK practitioners (56/95, 59%). But the differences by specialty or country were not significantly different (*p* > 0.05).

Participants were asked after how long they would consider switching the initially prescribed APD to another, and the majority of respondents reported that they usually decide to switch APDs after ‘4–6 weeks’ (102/190, 53.7%), with 59% of those from India, followed by ‘6–8 weeks’ (50/190, 26.3%), with 64% of those from the UK. They also switched after ‘3–6 months’ (22/190, 11.6%); the remaining (19/190, 10%) selected ‘others’ (6 from India and 13 from the UK). The timing for switching APDs between countries was significant (*p* < 0.05). The identified codes and themes from the participants’ open comments were switching due to poor efficacy (17/29, 59%), poor tolerance (9/29, 31%) and poor adherence to oral therapy (2/29, 7%); and one person (3%) said ‘not applicable’.

### 3.2. Antipsychotic Agent Switching or Combining

Of the reasons for switching a patient to another APD, the most selected reason was poor efficacy (130, 68%), followed by side effects (86, 45%), poor adherence (23, 12%) and patient request (23, 12%); and one participant selected pressure from nursing staff (0.5%). Side effects were the most selected option by the UK practitioners (86, 91%), and poor efficacy was the most selected option by the Indian practitioners (90, 95%). The reasons for switching APDs by country were significantly different (*p* = 0.004).

Reasons for consideration of prescribing a second APD included: 28.4% (54/190) reported that they consider concomitant treatment after 4–6 weeks, 24.2% (46/190) said after 3–6 months, 17.4% (33/190) said after 6–8 weeks, and 2.6% (5/190) selected never. The ‘after 4–6 weeks’ option was the most selected by the Indian practitioners (49/95, 52%). The ‘others’ option was allowed as a single option or in addition to another option from the given list. It was selected by 29% (55/190), 31% (17/95) from India and 69% (38/95) from the UK. The period before adding a concomitant drug to the initial APD by country was significant (*p* < 0.001).

Some participants (52/190, 27.4%) selected the ‘other’ option. Two identified themes included: to achieve a response (when switching APDs due to ‘lack of response)’ (31/52,47%) and ‘to improve response or reduce side effects’ (25/52, 38%). The rest (10/52, 15%) said ‘never’.

Poor efficacy was the most important reason for adding another APD (124/190, 65%), followed by side effects (23/190, 12%). The reason for adding a concomitant drug to the initial APD by country was significantly different (*p* = 0.014). The ’other’ option was allowed as a single option or in addition to one other option from the given list. This question was answered by few participants (53/190, 23%): 39 from the UK (73.6%) and 14 from India (26.4%). The top three identified themes were to improve patient outcomes (18/53, 34%), to achieve a response (17/53, 32%), not evidence-based (16/49, 30%), overlap during switching over (1/53, 2%) and nursing pressure (1/53, 2%).

### 3.3. Factors Leading to Patient Relapse

Participants listed the main factor leading to relapse in patients continuing therapy. Of 628 selections, participants selected patient non-adherence (153/628, 24.4%), followed by patient choice to cease therapy (126/628, 20.1%), concomitant use of illicit drugs (99/628, 15.8%), disease severity (84/628, 13.4%), concomitant use of alcohol (57/628, 9.1%), stress (56/628, 8.9%) and poor initial response (53/628, 8.4%). Fifteen respondents selected the option for ‘other’ (8/15, 53% UK and 7/15,47% India). The three identified themes were patient-related (13/15, 87%), medication-related (7/15, 47%) and practitioner-related (3/15, 20%). Concerning factors leading to repeated hospitalisations, the majority, both in the UK (85/95, 90%) and India (66/95, 70%), perceived patients’ lack of adherence as the main factor precipitating hospitalisations. The ‘other’ option was selected by 6% (11/190, 8, 73% UK and 3, 27% India). The identified themes were patient-related (9/11, 83.4%), practitioner-related (1/11, 8.3%) and medication-related (1/11, 8.3%).

For factors relating to relapse for patients no longer on APDs, of 431 responses, use of illicit drugs was the most selected option (119/431, 27.6%), followed by disease severity (114/431, 26.4%), stress (94/431, 21.6%), high alcohol consumption (56/431, 13.1%) and poor initial response (48/431, 11.3%). When the data were analysed by country of practice, the UK practitioners selected the use of illicit drugs (67/95, 71%) as the leading cause of relapse, and disease severity was mostly selected by practitioners from India (58/95, 61%). The ‘other’ option was selected by 12% (22/190, 15, 68% UK and 7, 32% India). Only two themes were identified: patient-related (17/22, 78%) and condition-related (5/22, 22%).

### 3.4. Use of Long-Acting Injectable Antipsychotic Formulations

Responding to reasons for prescribing long-acting injectable (LAI) formulations rather than oral forms, out of 440 total responses, the most common reason for prescribing LAI formulations was poor adherence (175/440, 39.8%), followed by the need for supervised dosing (94/440, 21.4%), patient request (61/440, 13.9%), difficult behaviour (42/440, 9.5%), community/court order (33/440, 7.5%), poor efficacy (20/440, 4.5%) and side effects (15/440, 3.4%).

On the acceptability of LAI formulations for patients, of 387 selections, most psychiatrists reported that schizophrenic patients were likely to accept LAI APDs administration over oral formulation (104/190, 55%). The practitioners selecting ‘unlikely’ were from the UK (54/95, 57%), and most of those who selected likely were from India (92/95, 97%). From a total of 387 selections, the most selected reasons for patient discontinuation from LAI APD formulations were side effects, followed by poor adherence to attending outpatient clinics (108/387, 27.9% and 101/387, 26.1%, respectively), followed by patient request (68/387, 17.6%), need for supervised dosing ended (45/387, 11.6%), community/court order ended (24/387, 6.2%), poor efficacy (23/387, 5.9%) and difficult behaviour control (18/387, 4.7%). The ‘other’ option was selected by 16% of all participants (24/190, UK 12/24, 50% and India 12/24, 50%). Themes were ‘medication related’ and ‘patient related’ (12/24, 50% and 8/24, 33%, respectively), followed by prescriber-related (3/24, 13%) and relapse (1/24, 4%).

### 3.5. Side Effects

The reported frequencies of side effects leading to poor patient adherence or the psychiatrists’ switching of an initial antipsychotic to another one or to stopping APD are summarised in Table 2. Weight gain was the most selected reason for non-adherence (10.8%) and for switching to another APD (8.1%), where life-threatening side effects (115/1091, 10.5%) and QT interval prolongation (102/1091, 9.3%) were the most selected reasons for stopping the use of APDs.

Concerning management of side effects, 56.3% (107/190) of psychiatrists responded that they would prescribe concomitant medications to manage side effects. There were 11 medications named 108 times: aripiprazole (70/108, 65%), clozapine (12/108, 11%), amisulpiride (9/108, 8%), risperidone (6/108, 6%), olanzapine (3/108, 3%), quetiapine (2/108, 2%), haloperidol (2/108, 2%), zuclopenthixol (1/108, 1%), ziprasidone (1/108, 1%), trifluperazine (1/108, 1%), cariprazine (1/108, 1%) and second-generation APDs in general (3/108, 3%).

### 3.6. Use of Scales for Assessment/Monitoring

About the scales used for assessing and monitoring patient symptoms, of 348 selections made, PANSS (The Positive and Negative Syndrome Scale) was the most used (122/348, 35.1%), followed by BPRS (Brief Psychiatric Rating Scale) (89/348, 25.6%), then CGI (Clinical Global Impression) (75/348, 21.6%), but YMRS (The Young Mania Rating Scale) was used the least used (62/348, 17.8%). Both Indian and UK practitioners selected PANSS as the preferred scale for assessing patients’ improvement (50/95, 53% and 72/95, 76%, respectively). There were 12% (22/190, India 9/22, 4% and UK 13/22, 7%) of respondents who selected the ‘other’ option; 50% (11/22) indicated that they use professional opinion only, 36.4% (8/22) selected local alternative validated scales, and one person (4.5%) selected visual analogue scale.

Practitioners were then asked about their perceived most common challenge they faced in managing psychotic disorders, and 86% (164/190) responded. They reported poor adherence (95/190, 58%), followed by lack of insight and knowledge (42/190, 26%), medications’ side effects (27/190, 17%), controlling symptoms (26/190, 16%), concomitant substance and alcohol use (11/190, 7%), controlling aggression (7/190, 4%), lack of resources and stigma equally at 3% (5/190), lack of social support (3/190, 2%), relapse (2/190, 1%), incorrect diagnosis and lack of professional empathy (equally at 1/190, 0.6%).

## 4. Discussion

### 4.1. Most Prescribed Medications

Patients’ individualised response (symptom control and tolerance) was highlighted in a number of systematic reviews [29,30,31,32,33,34,35]. In this survey, the most selected APDs were olanzapine (47.5%), risperidone (42.8%) and aripiprazole (25.3%), indicating that second-generation or atypical antipsychotics (SGAs) were the most prescribed APDs in the UK and in India in 2022. A survey conducted in India by Grover and Avasthi [36] showed that the top three anti-psychotics prescribed were risperidone 30%, olanzapine 30% and haloperidol 10.9%, with first-generation or typical antipsychotics (FGAs) comprising about 25.2% of all prescriptions in 22.4% of the cases. This survey suggests reduced usage of first-generation APDs and little support for the use of combinations. From the narrative responses, some of the differences between the two countries appear to reflect greater attention to the cost of therapy for the patient in India. In the UK, other than in private practice, the cost to the patient of most medicines available under the NHS is the standard prescription item charge of GBP 9.65 per item, or free to those under 16 years, those 16–18 years old in full-time education and anyone over 60 years, and there are cost ceilings for long-term therapy [37]. Zhang et al. [33] reported that olanzapine, amisulpride, risperidone and quetiapine were more effective and provided longer remission and less EPS than FGAs.

### 4.2. Reported Side Effects

In this survey, the most common troublesome side effects causing poor adherence were weight gain (10.8%), drowsiness (10.4%), erectile dysfunction and movement disorders (equally at 8.7%). Weight gain (8.1%), movement disorders (7.7%) and hyperprolactinaemia (7%) were the most reported side effects that caused psychiatrist to switch to another APD. Weight gain (11.4%) was the most common side effect reported by patients, prompting them to seek termination of their treatment, followed by drowsiness (10.3%) and erectile dysfunction (9.4%). In this survey, psychiatrists stopped APDs altogether mainly due to life-threatening side effects (10.5%), particularly QT interval prolongation (9.3%). Read and Williams [38] surveyed APD users, with 832 participants from 30 countries, mostly from the U.S., the UK and Australia. Of all participants, 56% felt their medication effectively addressed the issues for which they were prescribed, and 27% felt that the drugs made their problems worse. Some 41% found their medications helpful, whereas 43% found them unhelpful. Some 35% reported an improvement in their quality of life, whereas 54% reported a decline. The average number of adverse effects reported was 11, with an average of 5 being severe. Fourteen side effects were reported by over 57% of participants, including drowsiness, tiredness, sedation (92%), loss of motivation (86%), slowed thoughts (86%),and emotional numbing (85%). Additionally, 58% reported suicidality as a side effect. Older participants reported particularly poor outcomes and a high number of adverse effects. The duration of treatment was not related to positive outcomes but was associated with negative outcomes; 70% of participants attempted to stop taking their medications, with the most common reasons being the side effects (64%), concerns about long-term health (52%) and not being informed about side effects (70%) [38]. Weight increase with olanzapine, risperidone and clozapine and metabolic changes with olanzapine were greater [31]. Kishimoto et al. [32] reported that SGAs are recommended for maintenance treatment more so than FGAs in schizophrenia (*n* = 59 studies), but they did not report any in-class superiority. For all-cause discontinuation, clozapine, olanzapine and risperidone were significantly (*p* < 0.05) superior (less) to several other SGAs. Regarding intolerability-related discontinuation, risperidone was superior, and clozapine was inferior to several other SGAs. Olanzapine was worse for weight gain than all other SGAs and was superior to risperidone for EPS but not akathisia. Clozapine and quetiapine were significantly worse for sedation and somnolence than some other SGAs. Zhu [35] found that olanzapine was associated with more EPS, and quetiapine with less akathisia than haloperidol, aripiprazole, risperidone and olanzapine.

### 4.3. Reasons for Switching, Adding or Stopping APDs

In this survey, the majority of clinicians in India (60%) and the UK (48%) selected medication efficacy for the individual patient as the main reason for choosing a specific APD. The reasons for switching APDs were primarily poor clinical efficacy (68%) or lack of tolerability (45%). In the UK, side effects were the main reason (91%), and in India, it was poor clinical efficacy (95%). The majority of psychiatrists (53.7%) switched APDs to an alternative APD after 4–6 weeks, as opposed to 11.6% who would wait 3–6 months, with UK psychiatrists more likely to wait longer than their Indian colleagues before switching. The most reported reason for trying a second APD was poor efficacy (65%), with overlapping therapy often prescribed to cover the removal of the previous agent, but ongoing combined therapy was rarely used. Rare, life-threatening side effects were the main reason indicated as the reason to discontinue the use of APDs (10.5%), and the use of concomitant medications to manage side effects was mentioned by 56.3% as a means to prevent stopping or switching to another APD. Yen et al. [39] conducted a survey among psychiatrists in Taiwan regarding the discontinuation of APDs for patients who have recovered from first-episode psychosis. Of all respondents, 50.8% believed that only one in five patients would stop taking medication, whereas 34.7% believed nearer to two in five would stop. Only 1.7% believed that none of the patients would stop. The opinions on the period of observation before discontinuation varied, with 28.8%, 26.3% and 29.7% believing it should be <1 year, 1 year and 1–2 years, respectively. However, 15.2% suggested it should be longer. Of respondents, 27.1% thought discontinuation could be done within 6 months, 29.7% believed it could be done within 6–12 months, and the remainder believed it should be longer than 1 year. Of all psychiatrists, 37.3% believed that 40–60% of their patients would discontinue medication on their own, and more than 75% felt that 60% or more of their patients would consider discontinuing if given the opportunity [39]. Overall, as in this study, psychiatrists seem mostly to initiate change due to lack of efficacy, whereas patients request change due to side effects. Huhn et al. [29] reported that the use of APDs has an impact on patients’ morbidity, and as a result, adherence to therapy in which older APDs were used was often associated with more EPS and prolactin elevation, whereas many newer APDs produced more weight gain and sedation. The authors also added that this finding is important to note, as older ADPs are the most used in low-income and middle-income countries, where SGAs might not be affordable. Leucht et al. [28] reported that APDs differed substantially in the individual patient experience with side effects, and to a lesser extent in terms of efficacy. The authors disputed the correct classification of APDs as FGA and SGA and argued that clinicians should adapt the choice of APD to the needs of individual patients. Additionally, Hartling et al. [29] found a clinically important benefit of haloperidol over olanzapine for improving positive symptoms, but the benefit was scale-dependent. Similarly, they reported a clinically important benefit of olanzapine over haloperidol in improving negative symptoms when the PANSS was used but no difference in mortality for chlorpromazine vs. clozapine or haloperidol vs. aripiprazole. Incidence of the metabolic syndrome for olanzapine vs. haloperidol was 2% and 22%, respectively, and there was a higher incidence of tardive dyskinesia for chlorpromazine vs. clozapine (5% and 9%, respectively). Davis et al. [32] reported that clozapine, amisulpride, risperidone and olanzapine are superior to FGAs (*p* < 0.001). Zhu [33] identified 19 relevant randomised controlled trials of 12 antipsychotic drugs that involved 2669 participants. The authors reported that for the overall reduction of symptoms, amisulpride, olanzapine, ziprasidone and risperidone were significantly more efficacious than haloperidol. Olanzapine was superior to haloperidol and risperidone for the reduction of negative symptoms. Several SGAs were superior to haloperidol in terms of all-cause discontinuation.

### 4.4. Reasons for Switching to LAIs

In this survey, poor adherence was the main reason to use LAI over oral formulations. However, the main reasons for discontinuing LAIs were side effects that led to poor adherence to attend outpatient clinics (27.9% vs. 26.1%, respectively). Both UK and Indian psychiatrists reported that poor adherence was their main reason to use LAI over oral formulations (94% vs. 91%, respectively). In India, LAIs were less often used for treating patients because they are expensive for the patient, even though avoiding LAIs could lead to association with more non-adherence and relapse—and ultimately, higher overall costs of treatment. Hatano et al. [40], in a survey on patients’ satisfaction with APD formulations in Japan, reported that the most prevalent form of APD among schizophrenic patients (57.8%) was tablets, followed by LAIs (30.7%). Less than 10% of patients utilized powder, liquid or sublingual tablet formulations. The authors also reported that the main reason for high satisfaction with LAIs was ’did not forget medication’ (23.9%). However, ‘easy to take’ was the highest selected option for tablets (31.2%), orally disintegrating tablets (30.9%), powder (35.7%) and liquid (15.4%). For sublingual tablets, the highest scored option was ‘immediate effectiveness’ (16.7%). The main reason for dissatisfaction with LAIs was ‘injection site pain’ (also found in this study); for tablets, it was the ‘size of the tablets’ (12.6%); and for oral dispersible tablets (5.9%), liquid (23.1%) and sublingual tablets (16.7%), it was ‘unpleasant taste’. There were no significant differences observed among all formulations regarding satisfaction on the Drug Attitude Inventory-10 (DAI-10) scores (*p* > 0.05). Gundugurti et al. [41] surveyed psychiatrists in India on oral vs. LAI use regarding social functioning. They found that oral second-generation antipsychotic tablets were more commonly prescribed in acute treatment, whereas in the chronic phase, patients were treated with either tablets or LAIs. The study found that the use of LAIs resulted in lower relapse rates than tablets (12% vs. 60%), but as noted above and in this study, cost to the patient (in India) due to side effects and injection pain can affect ongoing adherence to therapy.

### 4.5. Possible Causes for Relapse

In this survey, the most prominent reason for relapse in patients was nonadherence (UK 90% vs. India 70%). Illicit substance use (27.6%) was the primary causes of relapse among patients who had discontinued or were not currently receiving APDs. Eisner et al. [42], in a thematic analysis of data from interviewing service users about interventions to prevent relapse in psychosis, created three themes as follows: identifying risk factors, early signs and reaction to deterioration. The subthemes of recognising risk factors were linking risk factors to relapse and reacting to risk factors. They also reported subthemes for identifying early signs were recognising early signs and recalling early signs. Their subthemes for reacting to deterioration were finding meaning in an overwhelming experience, coping with unusual experiences, help seeking and barriers to help seeking. The finding suggested most participants remembered some of the early signs of deterioration. Participants’ attention in recognising early signs of relapse was varied, but some participants reflected on recognising early signs through exacerbation of the voices that they heard. On the other hand, some mentioned they lacked insight. Some of the reported reasons to hinder the recall of their early signs were the existence of psychotic symptoms, cognitive problems or a ‘sealing over’ type of recovery. The speed of worsening of symptoms also affected the detection of early signs. In addition, inability to access clinicians at the point of relapse was considered a barrier in receiving help.

Gupta et al. [43] compared patients diagnosed with schizophrenia and treated with LAIs. Patients were regularly screened for substance abuse during therapy. They concluded that substance abusers had a significantly higher readmission rate to the hospital (mean of 2.5 admissions) compared to those who did not abuse substances (mean of 0.5 admissions), which was a statistically significant difference (*p* < 0.001) This again aligns with the experience of the psychiatrists in this survey, i.e., that ongoing substance use is a major negative indicator.

MacE and Taylor’s [44] survey on the prescription of SGAs for hospital in-patients was conducted among psychiatrists in England and Wales. Of participants, 28.6% used both FGAs and SGAs, and 19.3% received high-dose APDs. The co-prescription of both drugs was more common in patients > 40 years (32.0% vs. 25.3%, *p* = 0.018) and in centres employing senior pharmacists (28.6% vs. 14.3%, *p* = 0.03). High-dose APDs were more frequently prescribed to white patients (20.6% vs. 13.9%, *p* = 0.02) and patients > 40 years (24.4% vs. 15.0%, *p* < 0.001). The prescription of anticholinergics was significantly higher for patients receiving a combination of FGAs and SGAs compared to those taking only SGAs (26.0% vs. 12.0%, *p* < 0.001). Similarly, in this survey, the majority of psychiatrists stated that the primary reason for using a second APD was poor treatment efficacy (65%).

## 5. Limitations

This was a pragmatic study without external funding, undertaken during the global COVID-19 pandemic and lockdowns. The study is limited by the total sample reached. The cohort’s cross-sectional sampling of specialists available through publicly published contact addresses is also a weakness. This survey did not explore the effects of the psychiatrists’ traditional, cultural or religious beliefs, as this would have required a much longer and more detailed questionnaire, imposing a much greater time burden upon participants.

## 6. Conclusions

The most selected APD in both countries was olanzapine, followed by risperidone and aripiprazole. The majority of clinicians in India and the UK stated that medication efficacy for the individual patient was the main reason for choosing a specific agent. The main reasons for switching APDs were poor clinical efficacy and lack of tolerability. Switching of APDs was primarily for poor efficacy. The primary rationale for the administration of a second APD was poor treatment efficacy. The most prominent reason for relapse in patients undergoing treatment with APDs was nonadherence, with illicit substance use being the primary causes of relapse among patients who had discontinued or were not currently receiving APDs. The most troublesome side effects reported by psychiatrists were weight gain and drowsiness. Similarly, weight gain was the major side effect reported by patients as prompting them to seek termination of the treatment. Life-threatening rare side effects were the main reason that psychiatrists indicated as a reason to discontinue the use of APDs, and concomitant medications were used to manage side effects and to prevent stopping or switching. A particular area of concern was that the tools and scales used to report on responses and tolerability were only used by 35% of participating psychiatrists, and the remaining participants depended on patients’ self-reporting ratings such as visual numerical scales. Monitoring and adjusting therapy should be facilitated by an objective reference scale.

## Figures and Tables

**Table 1 pharmacy-11-00162-t001:** List of antipsychotics available in the UK.

Oral First-Generation (Typical)	Oral Second-Generation (Atypical)
Benperidol Chlorpromazine Flupentixol Haloperidol Levomepromazine Pericyazine Perphenazine Pimozide Prochlorperazine Promazine Sulpiride Trifluoperazine Zuclopenthixol	Amisulpride Aripiprazole Cariprazine Clozapine Olanzapine Paliperidone Quetiapine Risperidone

**Table 2 pharmacy-11-00162-t002:** Consequences of antipsychotics’ side effects.

Side Effect	Nonadherence to APDs (*n* = 1406)	Switching to Another APD (*n* = 1473)	Cease the Use of APDs (*n* = 1091)
Agitation	46, 3.3%	52, 3.6%	36, 3.3%
Amenorrhea	51, 3.6%	52, 3.6%	29,2.7%
Confusion	15, 1.1%	31, 2.1%	37, 3.4%
Dizziness	64, 4.6%	39, 2.7%	15, 1.4%
Drowsiness	146, 10.4%	96, 6.6%	50, 4.6%
Dry mouth	50, 3.6%	20, 1.4%	8, 0.7%
Erectile dysfunction	122, 8.7%	83, 5.7%	56, 5.1%
Fatigue	66, 4.7%	35, 2.4%	24, 2.2%
Galactorrhoea	52, 3.7%	67, 4.6%	46, 4.2%
Gynaecomastia	35, 2.5%	41, 2.8%	39, 3.6%
Hyperglycaemia	30, 2.1%	59, 4.0%	42, 3.8%
Hyperprolactinaemia	57, 4.1%	103, 7.0%	58, 5.3%
Hypotension	18, 1.3%	28, 1.9%	27, 2.5%
Insomnia	16, 1.1%	14, 1.0%	13, 1.2%
Life-threatening	0, 0%	78, 5.3%	115, 10.5%
Movement disorders	122, 8.7%	112, 7.7%	76, 7.0%
Muscle rigidity	79, 5.6%	78, 5.3%	45, 4.1%
Parkinsonism	90, 6.4%	90, 6.1%	48, 4.4%
Postural hypotension	35, 2.5%	30, 2.0%	28, 2.6%
QT interval prolongation	28, 2.0%	84, 5.7%	102, 9.3%
Seizure	11, 0.8%	50, 3.4%	67, 6.1%
Tremor	95, 6.8%	59, 4.0%	30, 2.7%
Urinary retention	19, 1.4%	36, 2.5%	34, 3.1%
Vomiting	7, 0.5%	8, 0.5%	6, 0.5%
Weight gain	152, 10.8%	119, 8.1%	60, 5.5%

## Data Availability

All additional data is available on individual request from H.M.

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
