# Peer review of "Antipsychotic Use: Cross-Sectional Opinion Survey of Psychiatrists in India and United Kingdom"

_pharmacy, 2023, doi:10.3390/pharmacy11050162_

Round 1

Reviewer 1 Report

This original article addresses a subject of interest, describing the antipsychotic use from the point of view of the prescriber. The topic is relevant for the field as it shows the opinion of doctors in areas with different health systems, cultural and socioeconomic backgrounds.   

This type of survey offers an opportunity for identifying prescription patterns and targeting the efforts toward the improvement of patient care. 

            To create an informed starting point, it would be beneficial for the reader to have concise information about the approach each health system has toward psychotic patients. Also, the antipsychotic drug classes could be listed in the introduction.

In the discussions section, some references regarding the efficiency of the antipsychotic drugs (other than surveys) would reinforce the opinions expressed by doctors. Furthermore, there are several studies regarding the official reports of side effects in international databases that could be referenced for comparison.

Please outline better the limitations of the study. The generalization of results is difficult, based on the reduced number of responses.

The references are appropriate, but the number is insufficient.  

Author Response

This original article addresses a subject of interest, describing the antipsychotic use from the point of view of the prescriber. The topic is relevant for the field as it shows the opinion of doctors in areas with different health systems, cultural and socioeconomic backgrounds.   

  • Thank you.

This type of survey offers an opportunity for identifying prescription patterns and targeting the efforts toward the improvement of patient care. 

  • Thank you.

To create an informed starting point, it would be beneficial for the reader to have concise information about the approach each health system has toward psychotic patients.

  • Added

Also, the antipsychotic drug classes could be listed in the introduction.

  • Now included in the introduction as table 1 and additional 2 paragraphs under the table.

In the discussions section, some references regarding the efficiency of the antipsychotic drugs (other than surveys) would reinforce the opinions expressed by doctors.

  • New articles are now included.

Furthermore, there are several studies regarding the official reports of side effects in international databases that could be referenced for comparison.

  • Now included in the introduction as table 1 and additional 2 paragraphs under the table.

Please outline better the limitations of the study. The generalization of results is difficult, based on the reduced number of responses.

  • Included.

The references are appropriate, but the number is insufficient.  

  • New references added.

Reviewer 2 Report

Thank you for giving me the opportunity to read and comment a report “Antipsychotic Use: Cross Sectional Opinions Survey of 2 Psychiatrists in India and United Kingdom”, by Sherzad Qadir Z. et al.

In the reviewed manuscript, the use of APDs in the treatment of psychosis, is analyzed.

This is a potentially interesting report but at present it is not suitable for publication.

Major issues

Authors should include a statistical analysis section describing the methods used.

The authors did not include a section on the limitations of the study. For the manuscript to be published, these limitations, such as the low response rate of the selected psychiatrists, should be included.

The authors have not included the required information in the following sections: Author Contributions, Funding, Institutional Review Board Statement, Informed Consent Statement, Data Availability Statement and Conflicts of Interest

Minor issues

It would be appropriate to include the meaning of the acronym SPSS.

Tables 1, 2, and 3 are repetitive. It would be desirable for the authors to summarize these data in a single table.

The quality of the manuscript would be enhanced if the authors included a table with the complete contents of the survey conducted, at least as supplementary material.

Author Response

Thank you for giving me the opportunity to read and comment a report “Antipsychotic Use: Cross Sectional Opinions Survey of 2 Psychiatrists in India and United Kingdom”, by Sherzad Qadir Z. et al. In the reviewed manuscript, the use of APDs in the treatment of psychosis is analyzed. This is a potentially interesting report but at present it is not suitable for publication.

  • Thank you.

Authors should include a statistical analysis section describing the methods used.

The authors did not include a section on the limitations of the study.

  • Now included.

For the manuscript to be published, these limitations, such as the low response rate of the selected psychiatrists, should be included.

  • Now added.

The authors have not included the required information in the following sections: Author Contributions, Funding, Institutional Review Board Statement, Informed Consent Statement, Data Availability Statement and Conflicts of Interest

  • Now included in the title page. Ethics and consent emailed separately to the editor.

It would be appropriate to include the meaning of the acronym SPSS.

  • Corrected.

Tables 1, 2, and 3 are repetitive. It would be desirable for the authors to summarize these data in a single table.

  • Table are now merged.

The quality of the manuscript would be enhanced if the authors included a table with the complete contents of the survey conducted, at least as supplementary material.

  • Now included.

Reviewer 3 Report

Thank you for the opportunity to review this paper.

Comments:

Last paragraph of Introduction includes parts that belong to method section - i.e. choice of web survey.

Rewrite to just include the aim and move sections that belong to methods to appropriate place.

The Introduction does not clearly indicate what to expect in the paper as a whole.  Please be more specific and reorganize it in a more logical order.

Why were these exact two countries chosen for comparison? This should be clear after reading the Introduction. What are those differences in guidelines between countries that could influence the practice? Also, what about health systems diffeeneces, reimbursement, waiting times etc? What about cultural differences, are there any that could be important? This should be mentioned.

PLease check the study is reported based on STROBE or other relevant guidelines. E.g. information on study design is missing in methods section.

Was the survey validated or pre tested?

Conclusions does not need detail repetition of findigs with percentages etc. Please rewrite in a more concise way. try also to communicate main implications of the study and not just repeat the findings

Author Response

Last paragraph of Introduction includes parts that belong to method section - i.e. choice of web survey. Rewrite to just include the aim and move sections that belong to methods to appropriate place. The Introduction does not clearly indicate what to expect in the paper as a whole.  Please be more specific and reorganize it in a more logical order.

  • Introduction corrected and readability improved.

Why were these exact two countries chosen for comparison? This should be clear after reading the Introduction. What are those differences in guidelines between countries that could influence the practice? Also, what about health systems differences, reimbursement, waiting times etc? What about cultural differences, are there any that could be important? This should be mentioned.

  • Reasons and explanation added.

Please check the study is reported based on STROBE or other relevant guidelines. E.g. information on study design is missing in methods section.

  • Corrected

Was the survey validated or pretested?

  • The survey was pre-tested with a small group of psychiatrists (two from each country) prior to dissemination.

Conclusions does not need detail repetition of findings with percentages etc. Please rewrite in a more concise way. try also to communicate main implications of the study and not just repeat the findings.

  • Corrected.

Round 2

Reviewer 2 Report

The authors have modified the manuscript according to the recommendations and it is now suitable for publication.

Reviewer 3 Report

All comments were addressed